# Regulation of LncRNAs in Melanoma and Their Functional Roles in the Metastatic Process

**DOI:** 10.3390/cells11030577

**Published:** 2022-02-07

**Authors:** Marine Melixetian, Pier Giuseppe Pelicci, Luisa Lanfrancone

**Affiliations:** 1Department of Experimental Oncology, IEO, European Institute of Oncology IRCCS, 20139 Milan, Italy; marine.meliksetyan@ieo.it (M.M.); piergiuseppe.pelicci@ieo.it (P.G.P.); 2Department of Oncology and Hemato-Oncology, University of Milan, 20122 Milan, Italy

**Keywords:** lncRNAs, melanoma, phenotype switch, metastasis, drug targeting, drug resistance

## Abstract

Long non-coding RNAs (lncRNAs) are key regulators of numerous intracellular processes leading to tumorigenesis. They are frequently deregulated in cancer, functioning as oncogenes or tumor suppressors. As they act through multiple mechanisms, it is not surprising that they may exert dual functions in the same tumor. In melanoma, a highly invasive and metastatic tumor with the propensity to rapidly develop drug resistance, lncRNAs play different roles in: (i) guiding the phenotype switch and leading to metastasis formation; (ii) predicting the response of melanoma patients to immunotherapy; (iii) triggering adaptive responses to therapy and acquisition of drug resistance phenotypes. In this review we summarize the most recent findings on the lncRNAs involved in melanoma growth and spreading to distant sites, focusing on their role as biomarkers for disease diagnosis and patient prognosis, or targets for novel therapeutic approaches.

## 1. Introduction

Melanoma is a malignant skin cancer arising from melanocytes, the pigment-forming cells of the skin [1]. It is highly prone to metastatic dissemination and recurrence after treatment, accounting for 73% of skin-cancer-related deaths in advanced-stage disease [2]. Early diagnosis of melanoma is crucial for patient prognosis and survival, as the 5-year survival rate for localized, early-stage melanoma is 99%, dropping to 65% for regional disease and only 27% for metastatic melanomas [3].

Melanoma has the highest mutational burden among cancers, partially attributed to UV-induced DNA damage, and is classified into four genetic subtypes: (i) the B-Raf proto-oncogene serine/threonine-kinase (BRAF) subtype (50%), defined by BRAF somatic mutations, targeting the V600 amino acid residue (V600E-V600K-V600R) or the K601 residue (K6001I); (ii) the Ras proto-oncogene GTPase (RAS) subtype (25%), including known damaging amino acid substitutions in all three RAS family members (NRAS, KRAS, and HRAS), more frequently NRAS; (iii) the Neurofibromin 1 (NF1) subtype (15%), defined by the presence of inactivating mutations in the NF1 negative regulator of the Ras signal transduction pathway; and (iv) Triple Wild-Type Subtype (10%), described as a heterogeneous subgroup characterized by a lack of hot-spot BRAF, N/K/HRAS, or NF1 mutations. Several COSMIC mutations were found in this subtype, including known drivers of uveal melanoma KIT, GNAQ, and GNA11, as well as CTNNB1 and EZH2 [4].

Melanomagenesis is a complex process involving sequential accumulation of somatic mutations, as reported in other cancers [5,6]. Mutations leading to the constitutive activation of the MAPK signaling pathway are considered as the putative sustaining-lesion as their targeting reverts or attenuates the melanoma phenotype [7]. Notably, BRAF or NRAS mutations are frequently mutually exclusive [8,9,10,11]. The BRAF V600E mutation is found in up to 80% of benign naevi, suggesting that it also functions as a cancer-initiating mutation [12]. By contrast, NRAS mutations or BRAF V600K or K601E mutations are more commonly associated with intermediate melanoma lesions, which had already accumulated other pathogenic mutations [9]. Progression from early to intermediate lesions and melanoma in situ is usually accompanied by the accumulation of mutations in the telomerase reverse-transcriptase (*TERT*) promoter and a high mutational burden [5,6,9]. TERT promoter mutations contribute to aberrant telomerase activity contributing to uncontrolled proliferation and immortalization of melanoma cells [6]. The acquisition of the invasive potential correlates instead with the accumulation of mutations affecting a distinct set of genes, including cell-cycle controlling genes (cyclin-dependent kinase-inhibitor 2A, CDKN2A), chromatin-remodeling genes (AT-rich interaction domain ARID1A, ARID1B, ARID2), and genes encoding SWI/SNF subunits. Finally, metastatic melanoma progression is often associated with inactivating mutations in the tumor suppressors phosphatase-and-tensin homolog (PTEN) or p53 (TP53) genes [9]). Melanoma metastases are also characterized by a further increase of the tumor mutational burden and copy number alterations, mainly affecting MAPK, phosphoinositide-3-kinase (PI3K), protein-kinase-B (AKT), mammalian-target-of-rapamycin (mTOR), Janus-kinase (JAK), and signal-transducer-and-activator-of-transcription (STAT) pathways [5,9].

Melanoma progression, however, does not always follow a defined evolutionary trajectory of driver gene mutations accumulation and consistently involves non-genetic alterations, including transcriptional and translational reprogramming, due to cellular responses to microenvironmental signals [13]. Indeed, metastasis dissemination involves complex interplays among melanoma cells, host intrinsic factors, and stromal cell populations. Consistently, malignant melanoma is endowed with high levels of intra-tumor heterogeneity and intrinsic plasticity, which enable melanoma cells to switch phenotypes and adapt to the changing and challenging surrounding environment [14].

Transcriptional reprogramming is a critical determinant of the phenotype switch of melanoma cells and is controlled by distinct transcriptional master-regulators, such as MITF, a crucial regulator of the melanoma phenotypic states [15,16]. Recent studies using single-cell RNA-sequencing technologies have allowed a detailed description of different phenotype-related transcriptional states [14] and uncovered the importance of phenotype plasticity in the development of metastasis, as well as drug resistance to both targeted- and immune therapy [17,18].

Despite numerous recent advances, melanoma progression and drug resistance still suffer from incomplete knowledge of the underlying mechanisms, representing the leading cause of mortality for this disease. Noncoding RNAs (ncRNAs) are emerging as critical regulators of tumor development in virtually all the analyzed tumor types [19]. Understanding their role in melanoma progression might offer new insights into the process of metastasis formation to better design new therapy approaches.

Noncoding RNAs are transcripts containing short, not evolutionary-conserved open reading frames (ORFs), usually not translated into functional proteins [20]. According to the human GENCODE database, the human genome contains about 16,000 ncRNA-transcribing genes, including snoRNAs, microRNAs, siRNAs, snRNAs, exRNAs, piRNAs, long ncRNAs, and the highly-abundant and functionally defined transfer RNA (tRNA) and ribosomal RNA (rRNA) [21].

Within the ncRNAs, long non-coding RNAs (lncRNAs) are longer than 200 nucleotides, are usually transcribed by RNA polymerase II from specific promoter regions, capped at the 5′ end, spliced, and frequently also polyadenylated at the 3′ end. Compared to protein-coding genes, lncRNA genes lack long ORFs, contain fewer exons (~2.8 as compared to 11 for protein-coding genes), and are usually expressed at very low levels in a tissue-specific manner. LncRNAs may contain specific sequence motifs to recruit specific nuclear factors that promote their nuclear and/or nucleolar localization [22]. Although, the nuclear localization of lncRNAs is tightly and coordinately regulated at different levels, from transcription to nuclear import/export; however, the mechanisms that control different nuclear localization patterns are still largely unknown. The functional roles of the nuclear lncRNAs were found to require the interaction with RNA-binding proteins (RBPs) [22]. A large fraction of the lncRNAs are instead exported to the cytosol, possibly sharing the same processing and export pathways of mRNAs. Upon arrival in the cytoplasm, lncRNAs are either assigned to specific cellular organelles or remain cytosolic and associated with RBPs to determine the function, as it happens in the nucleus [22]. However, nearly half to 70% of cytoplasmic lncRNAs are found in polysome fractions [23,24]. The functions of ribosome-associated RNAs are not fully understood; ribosome binding regulates lncRNAs turnover through nonsense-mediated mRNA decay, and there is an increasing body of evidence that numerous lncRNAs are involved in the regulation of translational initiation and elongation [24,25].

LncRNAs exert important roles in gene expression regulation, acting at multiple levels and in different cellular compartments. The LncRNAs regulate chromatin architecture, recruitment of transcription machinery components, mRNA stability, and, in the cytoplasm, translation and posttranslational processes [26,27]. Nuclear lncRNAs perform these functions by different mechanisms, by acting, for example, as scaffolds that regulate chromatin architecture or recruit chromatin-modifying enzymes (XIST, HOTAIR, DLX6AS [28,29,30,31], regulating long-range chromatin interactions [32,33,34], or regulating transcription directly by forming R-loops or interfering with the Pol II machinery [35,36,37]. Cytoplasmic lncRNAs are frequently involved in regulating mRNA turnover by different mechanisms [38,39,40,41], including association with ribosomes and direct translation-regulation [42,43,44]. An increasing body of evidence suggests that lncRNAs can also be translated, giving rise to micropeptides encoded by small ORFs [45,46,47,48].

An emerging group of lncRNAs are the circular RNAs (circRNAs)–circular RNA molecules that arise from pre-mRNA back-splicing through covalent linking of 5′ and 3′ pre-mRNA termini. About 80% of circRNAs are exonic cytoplasmic circRNAs that can interact with proteins and other RNAs, acting as microRNA sponges, thus regulating transcription or translation. Also, circRNAs can be translated into micropeptides [49,50,51,52].

A growing body of evidence suggests that lncRNAs are aberrantly expressed and deregulated in cancer, functioning as tumor suppressors or oncogenes [19,53]. Given the multiple mechanisms of action known for lncRNAs, it is not surprising that the same lncRNAs may exert dual functions in cancer, acting as either tumor suppressors or oncogenes, depending on the cancer type (H19, BANCR, TINCR, MALAT, XIST) [44,54,55,56,57,58,59].

This review summarizes the most relevant findings on lncRNAs involved in melanoma-genesis and metastatic dissemination, focusing on their role as biomarkers for disease diagnosis and patient prognosis or targets for novel therapeutic approaches.

## 2. Oncogenic lncRNAs in Melanoma

In recent years, several large-scale transcriptomic and genomic studies have identified deregulated lncRNAs in melanomas [60,61]. In a study using a collection of 7256 RNA-sequencing libraries from 27 tissues and cancer types, 339 lncRNAs were associated with melanoma [62].

Given the high frequency of BRAF mutations in primary melanomas, several studies aimed at identifying lncRNAs regulated by the mutant BRAF kinase. Analyses of transcriptome remodeling following BRAF^V600E^ ectopic-expression in normal melanocytes, for example, Flockhart et al. identified 39 annotated lncRNAs and 70 novel, non-coding transcripts induced by mutant BRAF, including BANCR, a novel lncRNA differentially expressed in primary melanomas as compared to normal melanocytes [61]. BANCR silencing downregulates MAPK signaling inhibits tumor growth and migration by upregulating the chemokine CXCL11. Mechanistically, BANCR acts as a competing endogenous RNA (ceRNA) for miR-204 and is associated with poor prognosis in melanoma [61,63,64]. Another BRAF-induced lncRNA is the novel non-coding transcript SPRY4-IT1 (Sprouty RTK Signaling Antagonist 4-Intronic Transcript 1) [65,66], transcribed as an independent transcript from the intronic region of the *SPRY4* gene [65]. SPRY4-IT1 is predominantly localized in cytoplasmic polysomes or ribosomal clusters and is overexpressed in melanoma [66]. Its silencing modulates lipid metabolism, impairs cell proliferation and invasion, and induces apoptosis in melanoma cells [66,67]. Conversely, overexpression of SPRY4-IT1 in normal melanocytes promotes proliferation, multi-nucleation, and anchorage-independent growth. Notably, high levels of SPRY4-IT1 in the plasma of melanoma patients correlate with tumor stage and poor overall survival [68]. Finally, two lncRNAs regulated by the BRAF pathway-MIR31HG and RMEL3- promoted melanoma proliferation and in vivo tumor growth in preclinical models by preventing p16^INK4A^-dependent cellular senescence and stimulating MAPK and PI3K pathways, respectively [69,70,71]. Their expression is associated with poor outcomes in melanoma patients [70,72].

Two candidate lncRNAs potentially linked to melanocyte transformation, Taurine upregulated gene 1 (TUG1) and urothelial cancer-associated 1(UCA1), was recently identified as negative regulators of melanogenesis and UVB response in normal melanocytes. TUG1 is overexpressed in melanoma and promotes tumor growth and metastasis formation in model systems by sponging miR-129-5p and miR-29c-3p [73,74,75]. Likewise, UCA1 overexpression correlates with tumor stage in melanoma patients and promotes proliferation and invasion by regulating the miR-28-5p/HOXB3 and miR-507/FOXM1 axes [76,77]. In normal melanocytes, UCA1 expression is induced by UVB and negatively correlates with MITF expression [78]. Still, until now, no experimental validation has functionally linked TUG1 and UCA1 expression to melanocyte transformation and primary melanoma formation.

The vast majority of melanoma-associated lncRNAs function as miRNA decoys leading to the activation of the main pathways deregulated in cancer: MALAT1, a well-characterized lncRNA upregulated in different cancers and functioning as ceRNA, promotes the epithelial-to-mesenchymal transition (EMT) by sponging miR-22 and miR-183 [79,80]; MHENCR activates the PI3K-Akt pathway by sponging miR-425/489 [81]; FOXD3-AS1 activates the MAPK3K2 pathway by sponging miR-325 [82]; KNCQ1OT1 upregulates expression of the MET receptor tyrosine kinase by sponging miR-153 [83]; lncRNA-ATB inhibits the Hippo pathway by upregulating YAP1 expression though miR-5980-5p [84]; NORAD is upregulated in melanoma tissues, where it promotes invasion and metastasis formation through the miR-205-EGLN2 pathway [85]; ZFAS1 stimulates expression of the RAB9A Rab GTPAse family member, by sponging miR-150-5p [86].

Several lncRNAs aberrantly expressed in melanoma regulate gene transcription by acting as scaffolds for chromatin-modifying complexes. In particular, CASC15 and LNMAT1 repress transcription of the programmed cell death 4 PDCD4 and Cell Adhesion Molecule 1 CADM1 tumor suppressor genes, respectively, by recruiting polycomb repressor complexes to target gene-promoters. Transcriptional repression by PDCD4 and CADM1 leads to increased invasion and metastasis formation in melanoma cells [87,88,89,90]. The ANRIL lncRNA (antisense non-coding RNA transcribed from the INK4 locus) inhibits transcription of the INK4A and INK4B genes, induces transcriptional reprogramming through the recruitment of polycomb complexes, and possesses well-established oncogenic properties in melanoma [91,92,93]. The homeobox transcript antisense intergenic RNA (HOTAIR) is one of the 231 ncRNAs associated with the human HOX loci. Its expression in melanomas correlates with tumor stage and its presence in the plasma of melanoma patients with advanced disease [94,95]. HOTAIR is an epigenetic regulator that acts as a scaffold for histone-modification complexes [28,96]. In melanoma, HOTAIR also functions as ceRNA for miR-157-3p and activates the c-MET oncogene and the PI3K/AKT/mTOR-signaling pathway, promoting invasion and metastasis [95].

Newly achieved data suggests that lncRNAs control various metabolic pathways in melanoma. The SAMMSON lncRNA regulates mitochondrial metabolism and protein translation by interacting with p32, a master regulator of mitochondrial homeostasis and metabolism. The SAMMSON gene is co-amplified in melanomas together with MITF within the genomic locus 3p13-3p14 and is associated with poor prognosis in melanoma patients. Notably, SAMMSON expression is required for melanoma growth in vivo and in vitro and confers resistance to MAPK inhibitors [97,98]. The lncRNAs H19 and CCAT1, instead, control melanoma cell invasion in vitro and tumor growth in vivo by regulating glucose metabolism [99,100,101]. The lncRNA LINC00518 directly regulates levels of the hypoxia-inducible factor HIF1α and glucose consumption in melanoma cells and promotes melanoma invasion, tumor growth, and radio-resistance in mice xenografts [102]. Importantly, these lncRNAs display low expression in normal tissues and, for this reason, might represent promising drug targets and biomarkers in melanoma.

Natural antisense transcripts (NAT) constitute one of the most important lncRNA classes aberrantly expressed in melanoma. Oncogenic antisense lncRNA LHFPL3-AS1 is overexpressed in melanoma, particularly in the ALDH1-positive subpopulation of the MDA-MB-435 melanoma cell line, enriched of *bona fide* melanoma initiating cells [103]. LHFPL3-AS1 upregulates the miR-181a-5p/BCL2 and miR585/STAT3 feedback loops, which are required for melanoma migration, invasion, and growth in vitro and in vivo and is an unfavorable prognostic marker in melanoma patients [104,105]. Another natural antisense transcript, the lncRNA TTN-AS1, regulates in *cis* the activity of the Titin gene promoter and stimulates melanoma cell migration, invasion, and tumor formation in vivo [106]. The already mentioned OIP5-AS1 is a NAT transcribed from the promoter of the cancer-testis specific gene OIP5, is overexpressed in melanomas, and drives glutamine catabolism by targeting the miR-217-glutaminase negative feedback loop. Knockdown of OIP5-AS1 impairs in vivo tumor growth in a mouse xenograft model [107].

Circular RNAs are an emerging class of lncRNAs involved in cancer and immune regulation [108]. CircRNAs mainly act as ceRNAs for cancer-associated miRNAs, regulating melanoma growth, invasion, EMT, and immune evasion. The circular RNA circ_0016418 promotes glutamine catabolism by upregulating glutaminase expression [107,109]. To date, only circ_0084043, circ_0020710, and circ_0016418 have been demonstrated to function as oncogenes in melanoma in immunocompromised mice [109,110,111,112,113,114]. Given their low expression in normal tissues, several lncRNAs are regarded as promising melanoma biomarkers (reviewed in [115]) (Figure 1).

## 3. Tumor-Suppressor lncRNAs in Melanoma

Even though most lncRNAs are upregulated in melanoma compared to normal tissues, some are downregulated in primary and metastatic melanomas, suggesting a role in tumor suppression. The MITF and SOX10 transcription factors act as important regulators of melanoma phenotypic-state transition and control invasive and metastatic features of melanoma [116]. Thus, the lncRNAs targeted by the MITF-SOX10 network can also function as potential tumor suppressors and inhibitors of metastasis formation, yet their function is still largely uncharacterized. Coe et al. identified 245 lncRNAs targeted by MITF-SOX10. Among these, the authors identified and characterized a novel lncRNA and DIRC3, expressed in melanocytes and silenced in proliferating MITF-SOX10 high-expressing melanomas [117]. DIRC3 acts in *cis* as an epigenetic regulator of IGFBP5 gene expression and inhibits invasion and migration of melanoma cells.

The NFκB-interacting lncRNA (NKILA) is a well-characterized tumor suppressor in various cancer types [118,119,120,121]. In melanomas, NKILA inhibits invasion and metastasis dissemination [122] through the inhibition of NFκB signaling. However, Huang et al. identified NKILA expression as a factor contributing to immune evasion in breast cancer PDX models [123], suggesting a further function of NKILA in cancer growth.

A recent study identified the circular cerebellar degeneration related1 antisense transcript (CDR1as) as a novel potential tumor suppressor in melanoma. CDR1as is a splice product of the LINC00632 lncRNA transcript highly expressed in melanocytes and epigenetically silenced by EZH1/PRC2 in primary and metastatic melanomas. CDR1as interacts with the Insulin-like growth factor-binding protein 3 (IGF2BP3); its silencing activates IGF2BP3 and promotes melanoma invasion, resistance to GPX4 inhibitors, and tumor growth in vivo [124].

The lncRNA growth arrest-specific transcript 5 (GAS5) is a tumor suppressor in multiple cancers [125]. In melanoma, GAS5 inhibits growth and invasion by acting as a ceRNA for miR-137 and as a regulator of oxidative stress [126,127,128,129].

The lncRNA CPS1 Intronic Transcript 1 (CPS1-IT1) was recently described as a tumor suppressor in different cancer types [130,131,132]. In melanoma, loss of CPS1-IT1 expression correlates with metastasis formation and clinical stage. The expression of CPS1-IT1 in melanoma cells inhibits cell migration, invasion, EMT, and angiogenesis by repressing the expression of CYR61, a SMARCA4-dependent inducer of angiogenesis [133]. Other lncRNAs that inhibit melanoma growth and metastasis dissemination include LINC00961, LINC00459, and MEG3, all acting as miRNA decoys [134,135,136,137,138].

We identified TINCR lncRNA as a suppressor of melanoma invasion and metastasis formation [44]. TINCR expression prevents translational reprogramming, activation of ATF4, and stress response in melanoma. We showed that TINCR is mildly expressed in melanocytes and upregulated in primary melanomas, where it functions as a barrier to the invasive phenotype switch. Consistently, metastatic melanomas display very low levels of TINCR RNA. Restoration of high-level expression of TINCR in metastatic melanomas reverts the invasive phenotype, thus demonstrating the direct involvement of TINCR in the metastatic process [44]. Importantly, TINCR was described as an oncogene in other cancers, acting as ceRNA for various miRNAs. These data suggest that TINCR regulates several cancer pathways through different mechanisms and that the contribution of TINCR to cancer development is cell- and tissue- context-dependent (Figure 1).

## 4. LncRNAs in the Phenotype Switching and Metastasis Formation Process

Melanoma cells display high inter- and intra-tumor heterogeneity and phenotype plasticity, allowing them to adapt to hostile tumor microenvironmental (TME) conditions. The latter process, called “phenotype switch”, is a reversible process of transcriptional and metabolic reprogramming of melanoma cells that can transit from a melanocytic/proliferative to a mesenchymal/highly-invasive phenotype, passing through different intermediate states, and vice versa [116,139]. These phenotypic transitions are an important requisite for the dissemination of melanoma and subsequent metastatic niche formation [18]. Importantly, while many factors (cytokines, extracellular ligands, small molecule compounds, etc.) were identified as inducers of the invasive cell state, only a few were described as capable of inducing the reverse switch [18].

The phenotype switch is a complex process that involves the coordinated regulation of many different genes and signaling pathways. Recent transcriptomic studies revealed that lncRNAs might be engaged in regulating phenotype switches in melanoma. One is the lncRNA TINCR, whose ectopic expression reduces melanoma cell invasion and expression of invasive markers, driving the transition of invasive cells to intermediate and proliferative cell states [44,140]. TINCR is highly expressed in primary melanomas, is downregulated in metastatic melanomas, and is required to maintain the proliferative cell state. TINCR downregulation inhibits melanoma development in vivo by suppressing the ATF4-CHOP pathway [44] and activating the Hippo pathway [140]. However, the association of TINCR expression with melanoma clinic-pathological features and prognosis is still missing.

Another inducer of the melanoma phenotype switch is the antisense lncRNA ZEB1-AS1, a marker of poor prognosis in many cancers [141,142,143]. ZEB1-AS1 is transcribed from the same promoter of ZEB1 and positively regulates its expression either in *cis* or at the post-transcriptional level [144,145,146]. Given the critical role of ZEB1 in EMT transition in various tumor types and phenotype switch and drug resistance in melanoma [147,148], ZEB1-AS1 represents an attractive stratification biomarker and molecular target in melanoma.

A role in melanoma phenotype switch has been hypothesized for several other lncRNAs. CASC15 [88] is an oncogenic lncRNA associated with melanoma progression, which activates the Wnt/β-catenin pathway and inhibits expression of PDCD4 [87,88,89] by recruiting EZH2 and increasing H3K27me3 level at its promoter [87]. However, the effects of CASC15 silencing on the invasive phenotype, expression of mesenchymal markers, and transcriptional reprogramming are still controversial [87,88].

HOTAIR expression is highly increased in metastatic melanomas and also in the lymphocytes surrounding the metastatic cells, suggesting a prominent role of the lncRNA in the invasion process by sponging miR-152-3p to upregulate the tyrosine kinase c-MET, known to be involved in metastasis [94].

The CircRNA_0084043 was reported to be implicated in EMT through direct binding to miR-153-3p, a tumor suppressor capable of regulating EMT by targeting SNAIL (Luan et al., 2018a). Consistently, downregulation of either circRNA_0084043 or miR-153-3p significantly reduces both proliferation and migration of melanoma cells and the levels of SNAIL (Luan et al., 2018a). Furthermore, circ_0084043 expression positively regulates TRIB2 (a scaffold and atypical kinase that signals to canonical MAPKs and regulates the substrates’ ubiquitination) by sponging miR-429 Downregulation of TRIB2 upon circ_0084043 knockdown inhibits the Wnt/β-catenin signaling, thus suggesting a role of circ_0084043 in the phenotype switch in melanoma [149].

Finally, LADON, the natural antisense transcript of the TGFβ family member NODAL, promotes the mesenchymal to the amoeboid transition of melanoma cells, a crucial invasion step. Indeed, LADON downregulation allows melanoma cells to maintain a non-invasive phenotype, while its overexpression induces cell motility ad reorganization of the actin cytoskeleton and attenuates cell proliferation. LADON expression is dependent on WNT/b-Catenin signaling, whose activation inhibits the expression of the metastasis suppressor NDRG1 [150] (Figure 1).

## 5. LncRNAs and Immune Evasion in Melanoma

Melanoma progression depends not only on the accumulation of genetic alteration and the acquisition of new phenotypic states but also on the interaction of melanoma cells with the immune system [151]. The melanoma niche contains many immune-suppressive cells: regulatory T-cells, myeloid-derived suppressor cells, and tumor-associated macrophages [152]. During melanoma progression, a number of soluble factors released by either tumor or ‘mmune-suppressive cells inhibit the capacity of antigen-presenting cells, dendritic cells, or macrophages to process and correctly expose melanoma-associated new antigene, leading to impaired T-cell activation and immune evasion [153,154,155]. Recently, pan-cancer genome-wide studies identified lncRNAs associated gene-expression signatures of the pro-tumor and anti-tumor TME, associated with response to immunotherapy [156,157,158]. Notably, immune-related lncRNA signatures were shown to be predictive of prognosis, survival, and immunotherapeutic efficacy in breast cancer, lung adenocarcinoma, hepatocellular carcinoma, bladder cancer [159,160,161,162,163], as well in melanoma [157,158,163,164,165,166,167,168]. Four distinct lncRNAs immune-related signatures predictive of patient prognosis and survival were identified using the cutaneous skin melanoma (SKCM) TCGA dataset, including lncRNAs related to antigen processing and presentation, cytokines, Toll-like and cytokine receptor signaling pathway, and NK cell-mediated immunity [163,165,167,168].

In recent years, advances in understanding the cellular and molecular mechanisms of tumor immune-escape led to the development and approval of many immunotherapies. Immune checkpoint inhibitors (ICI) targeting programmed death-1 (PD-1), its ligand PD-L1, and cytotoxic T lymphocyte-associated antigen 4 (CTLA-4) have revolutionized the treatment of metastatic melanoma, leading to long-term remissions or cure. However, about 50–60% of patients still show primary resistance to ICI therapy or develop secondary resistance during treatment, and the underlying molecular mechanisms are not fully understood. Among the most characterized cell-autonomous resistance mechanisms is the downregulation of b2-microglobulin and human leukocyte antigens (HLA) expression, leading to tumor evasion from T-cell specific cytotoxicity. Primary ICI-resistance in melanoma is frequently associated with a low mutational burden [169,170,171,172].

Yu et al. identified a 4-lncRNA signature (AC002116-2, AP000251-1, TMEM147-AS1, and NKILA) associated with a response to immunotherapy (anti-PD-1, CTLA4 and a cytokine tumor vaccine in 71 melanoma patients from the SKCM TCGA dataset) [166]. In another study, Zhou and colleagues using two independent training and validation melanoma-datasets identified a 15-lncRNA signature (AC010904.2, LINC01126, AC012360.1, AC024933.1, AL442128.2, AC022211.4, AC022211.2, AC127496.5, NARF-AS1, AP000919.3, AP005329.2, AC023983.1, AC023983.2, AC139100.1, and AC012615.4) that predicts response to anti-PD-1 monotherapy [164] (Figure 1).

To date, the only lncRNA known to play a direct role in the anti-melanoma immune response is LIMIT, a lncRNA that induces MHC-I expression and tumor immunogenicity. The LIMIT gene is conserved in mice and humans, and its expression correlates with lymphocyte infiltration and expression of interferon response genes in the SKCM TCGA dataset. LIMIT expression is induced by IFNγ treatment in melanoma cells and *cis*-activates the guanylate binding protein (GBP) gene cluster, leading to activation of the MHC-I antigen presentation machinery through the heat shock factor 1 (HSF1). Li and colleagues showed that increased LIMIT expression in melanoma patients and mouse transplanted melanomas correlates with MHC-I response, antitumor immunity, and enhanced efficiency of anti-PD-1 therapy. This is the first discovery of an immunogenic lncRNA in cancer, whose therapeutic targeting can represent a useful approach to increase the efficacy of cancer immunotherapy [173].

Circular RNAs are an emerging class of RNAs involved in the regulation of immune response [174]. A recently described circRNA, circ_0020710, deriving from CD151 mRNA encoding an oncogenic transmembrane protein, promotes melanoma progression and contributes to melanoma immune evasion by sponging miR-370-3p and upregulating CXCL12 levels. It was shown that high levels of circ_0020710 correlate with cytotoxic T-lymphocyte exhaustion in melanoma patients and that combined treatment of the CXCL12 inhibitor AMD3100 and anti PD-(L)1 enhance its therapeutic efficacy in mouse xenograft melanoma models [114]. Recent findings support novel functions of cirRNA in natural anti-cancer immunity. The innate immune system in mammals can distinguish between native vs. exogenous circRNA. N6-methyladenosine (m6A) methylation is critical for discrimination, which is the most frequent and abundant transcriptional modification in eukaryotic RNAs. While m6A-modified circRNAs inhibit immune response and are recognized as self cirRNAs, foreign m6A-unmodified circRNAs are potent activators of the anti-tumor immune response, opening the possibility of using circRNAs in cancer immunotherapy [175]. Still, the role of m6A modification in anti-tumor immune response remains controversial. It was recently described that circNDUFB2 triggers an antitumor immune response in non-small lung cancer (NSCLC) mouse model independently of m6A modification [176].

Increasing evidence suggests that small ORFs in the lncRNAs are translated into small peptides, representing a source of cancer-associated antigens (neoantigens) in different cancer types [52,177]. Increased neoantigen load in cancer cells is reported to increase the efficiency of immune-checkpoint inhibitor (ICI) therapy [178,179]. Qi et al. performed comprehensive proteogenomic profiling of HLA class 1-presented immunopeptides in melanomas with a high tumor mutation burden. The authors identified 44 lncRNA-derived peptides presented by HLA class I. These data suggest that deregulation of lncRNA expression in melanoma can impact neoantigen load and predict the response of melanoma patients to immunotherapy (Figure 1).

## 6. LncRNAs and Drug Resistance in Melanoma

In recent years, the development of targeted small-molecule inhibitors has revolutionized melanoma standard-of-care and improved patient overall survival. The most frequently targeted cellular pathway in melanoma is the MAPK/ERK pathway, activated by the RAS and BRAF mutations in most melanoma patients. Both are gain-of-function mutations, driving MAPK pathway activation via MEK1/2 and ERK1/2 [180,181,182]. The discovery of the oncogenic BRAF mutation and the activation of the MAPK pathway signaling led to the development of successful combination therapies, targeting BRAF with either vemurafenib [183,184] or dabrafenib [185] and MEK kinase with trametinib [186]. Although clinical trials with MEK inhibitors have shown less impressive responses than BRAF inhibitors in monotherapy, combination therapies have significantly improved initial results showing unprecedented clinical responses [187].

However, intrinsic (5%) and acquired (50%) resistance to targeted therapy is still the major challenge for melanoma patients. Two major mechanisms underlie the onset of therapy resistance. The first is the genetic drug resistance which occurs via a selection of mutations inhibiting the response to selective, targeting inhibitors mainly through the acquisition of mutations in the drug-binding domain of targeted kinases and in the pathway downstream effectors [188]. The second mechanism relies on the intrinsic phenotypic plasticity and intratumor heterogeneity of melanoma cells which trigger adaptive responses to therapy and acquisition of drug resistance phenotypes [18]. The best-characterized mechanism is the phenotype switch from MITF^high/^AXL^low^ drug-sensitive to MITF^low/^AXL^high^ drug-resistant melanoma cells induced by inhibitors of the MAPK pathway [18,189,190]. Unfortunately, most patients develop secondary resistance and eventually relapse despite the initial responses. Emerging evidence suggests that the same lncRNAs involved in the phenotype switch in melanoma are also involved in the acquired resistance to MAPK pathway inhibitors.

The lncRNA TSLNC8 is downregulated in melanoma patient samples with acquired resistance to vemurafenib and vemurafenib-resistant melanoma cell lines. Mechanistically, TSLNC8 binds to and regulates the subcellular localization of the catalytic subunit of protein phosphatase 1α (PP1α). Knockdown of TSLNC8 results in the cytoplasmic accumulation of PP1α and re-activation of the MAPK signaling [191].

The lncRNA SAMMSON, which controls mitochondrial metabolism and functions as an oncogene in melanoma, is upregulated following inhibition of the ERK signaling in mutant BRAF melanoma cells, and its depletion sensitizes melanoma cells to BRAF inhibitors [97]. Conversely, ectopic expression of SAMMSON confers resistance of melanoma cells to vemurafenib [98].

The long intergenic lncRNA *MIRAT* is upregulated following prolonged MAPK inhibition in *NRAS* mutant melanomas and modulates MAPK signaling by binding to the MEK scaffold protein IQGAP1 [192]. The lncRNA POU3F3 promotes resistance of melanoma cells to the alkylating agent dacarbazine, upregulating the DNA-methyltransferase MGMT levels by sponging miR-650 [193]. Finally, the lncRNAs XIST, H19, MEG3, and LINC01291 were shown to confer resistance of melanoma cells to platinum compounds by regulating miR-21/PI3KR1 miR-18/IGF1, miR-499-5p/CYLD, and miR-625-5p/IGF-1R axis, respectively [136,194,195,196].

Recently, we discovered that the lncRNA TINCR plays a novel role in drug resistance in melanoma patients. TINCR knockdown leads to ATF4 activation, downregulation of MITF, and acquisition of an invasive phenotype, accompanied by increased resistance to vemurafenib and trametinib. Importantly, overexpression of TINCR in melanoma patient-derived xenografts partially reduced melanoma cell invasiveness and sensitized cells to the MEK1/2 inhibitor trametinib [44] (Figure 1).

## 7. LncRNAs as Biomarkers in Melanoma

LncRNAs can be secreted by cells and detected in different body fluids, such as blood, plasma/serum, or urine [197]. They originate from apoptotic and necrotic cells or are actively secreted by living cells through extracellular vesicles. Secreted vesicles protect lncRNAs from degradation by RNAses and are therefore good candidates as stable markers for diagnosis or prognostic stratification [198].

Several circulating lncRNAs have been described in melanoma patients. lncRNA HOTAIR was found in the plasma of patients with advanced melanoma, while HOTAIR expression in melanoma tumors strongly correlates with tumor stage [94]. The lncRNA LINC01638 is significantly upregulated in the plasma of melanoma patients and predicts local recurrence [199]. Similarly, SPRY4-IT1 expression is increased in the plasma of melanoma patients compared to healthy individuals, and its expression highly correlates with tumor site and stage [68]. The plasmacytoma variant translocation 1 (PVT1) lncRNA was also detected at elevated levels in the serum of melanoma patients, and its expression correlates with tumor stage and is a marker of postoperative disease dynamics [200] Kolenda and colleagues identified a 17-lncRNAs signature in the plasma of melanoma patients that distinguishes healthy individuals and melanoma patients. Three of these lncRNAs-IGF2AS, MEG3, ZEB2-AS1—were identified as independent prognostic factors in *BRAF*-mutant advanced melanoma patient sera treated with vemurafenib [201].

## 8. LncRNAs as Drug Targets

Given their function in cancer progression, lncRNAs represent attractive targets for developing new drugs. Different RNA-based approaches were employed to create lncRNA-targeting drugs, such as posttranscriptional downregulation using antisense oligonucleotides (ASOs), small interfering RNAs (siRNAs), small hairpin (shRNAs), therapeutic circular RNAs, and CRISPR-Cas9 gene editing approaches [202,203,204]. The last generation of ASOs includes phosphorothioate oligonucleotides and locked nucleic acids (LNA) modifications, chimeric RNA-DNA-RNA ASOS (GAPmers) bearing LNA, and S-constrained ethyl modifications that have demonstrated improved potency and in vivo stability [97,205].

A new approach to lncRNA targeting involves the steric inhibition of lncRNA interactions with nucleic acids and proteins or targeting a small molecule of lncRNA elements responsible for secondary and tertiary RNA structure formation. These small molecules can potentially destabilize lncRNA molecules or inhibit their biological functions by inhibiting their molecular interactions. The small molecule NP-C86 has been shown to disrupt the interaction of the tumor suppressor GAS5 lncRNA with UPF1, a protein involved in nonsense-mediated decay, leading to upregulation of GAS5 RNA [206]. Two other studies identified small molecules targeting the 3′ triple-helix element of MALAT1 lncRNA, leading to a significant reduction of its expression [207,208]

Natural Antisense RNAs have been used to *cis*-regulate neighboring genes, as shown for the CDKN2B and CDKN1A loci by the lncRNA ANRIL [91] and LincRNA-p21 [209], respectively. The long intergenic noncoding RNA p21 has also been described as an inducer of the transcriptional activity of wild-type p53 [210]. Currently, several chemically modified single-stranded ASOS, named antagoNATs, are under preclinical and clinical development [211,212].

In addition, recent advances in RNA-based therapeutics approaches and applications open novel possibilities for using lncRNAs as drug themselves [203,204]. The delivery system is key to properly exploiting the envisioned drug role for lncRNAs. To be satisfactory, lncRNAs should be conveyed with high stability, specificity, cell permeability, and low immunogenicity. The most frequently used carriers are viral vectors, both lentiviral and adenoviral vectors, which can rapidly infect dividing and non-dividing cells and have very long-expression time. However, the safety of viral vectors in systemic drug administration has been a matter of discussion, and these vectors have been recently replaced by non-viral vectors, such as liposomes, lipid nanoparticles (NPs), and exosomes [203].

Liposomes are an ideal type of nanoparticles for drug and DNA/RNA delivery, as they show reduced toxicity and immunogenicity, and encapsulation improves drug stability. Several liposome-encapsulated lncRNAs are used in preclinical experiments in vitro and in vivo [213,214,215].

Lipid NPs are similar to liposomes, but they are more suitable for encapsulating lncRNAs. They proved to inhibit tumor growth in preclinical animal trials, with no serious side effects [216]. Recently, ASO-gold-TAT NPs were used to target the nuclear lncRNA *MALAT1* in a mouse model of xenotransplanted lung cancer, showing the efficient suppression of metastasis dissemination and increased overall survival [217].

Exosomes are membrane-bound, endogenous vesicles containing lipids, proteins, DNA, mRNAs, miRNAs, and lncRNAs that are naturally secreted by cells [218]. Exosomes are very promising delivery vectors and can be easily engineered to convey lncRNAs. They show lower immunogenicity and higher stability in vivo but an average packaging efficiency compared to liposomes. For this reason, exosomes can now be integrated with liposomes and NPs to improve specificity and deliverability. Liposomes, lipid NPs, and exosomes can be functionalized and delivery optimized by surface modification through genetic engineering or chemical modifications. Vesicles can be coupled with nucleic acid aptamers, antibodies, peptides, protein ligands, polymers, or small molecules [203].

Even if several lncRNAs proved to be promising drug targets or drug themselves in both in vitro and in vivo models, up to now, none of the lncRNA or small molecule targeting lncRNAs have entered clinical trials. Moreover, the initial clinical evaluation of RNA-based drugs in cancer is still controversial: four drugs entered phase II or III clinical trials, but seven were withdrawn because of the off-target effects and lack of efficiency [203]. The delivery, specificity, and immunogenicity of RNA-based drugs remain the major challenges in developing non-coding RNA therapeutics.

## 9. Conclusions

Over the past decade, a growing number of studies have demonstrated the crucial role of lncRNAs as oncogenes and tumor suppressors, highlighting their importance in regulating different cancer hallmarks, thus leading to uncontrolled proliferation, metastasis, and immune escape [19,219].

LncRNAs exert their functions through multiple mechanisms, acting as ceRNAs, regulators of chromatin structure, transcription, mRNA metabolism, stability and splicing, translation, and protein stability. The functions of lncRNAs are cell-context dependent and tissue-specific, and growing evidence suggests that the same lncRNA can act as either tumor suppressor or oncogene depending on cell and tissue contexts [220,221]. These findings established the rationale to investigate lncRNAs as diagnostic/prognostic biomarkers and/or therapeutic targets. The current limit in the knowledge of lncRNAs in melanomagenesis is, however, their poor characterization in vivo, both in terms of their role in tumorigenesis and in physiological function, the latter critical for the design of drug development pipelines. Moreover, the mechanisms of lncRNAs functioning in anti-tumor immune response and tumor microenvironment represents a major hurdle, as they are still only partially understood, mostly because of the difficulties in establishing appropriate in vivo models (Figure 2).

From the translational point of view, lncRNAs are emerging as previously underestimated therapeutic and diagnostic targets. Given the restricted, cancer-specific expression of many oncogenic lncRNAs, their diagnostic testing as cancer biomarkers in liquid biopsies held great promise. Ongoing massive efforts in the field of RNA-therapeutics will allow the exploitation of the usage of lncRNA as efficient drugs in the modulation of melanoma growth and its interaction with the immune system.

## Figures and Tables

**Figure 1 cells-11-00577-f001:**
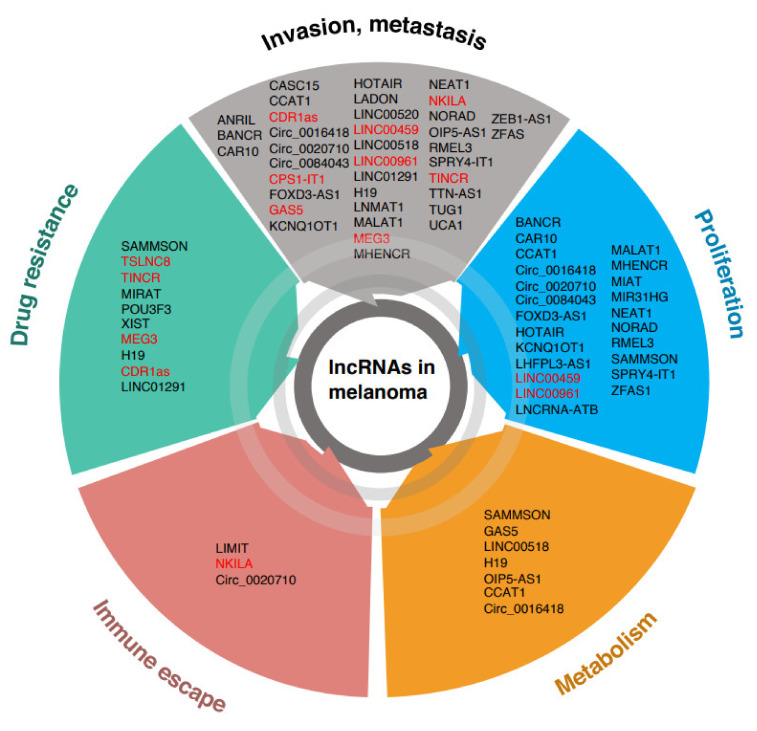
Overview of lncRNAs associated with major hallmarks of melanomagenesis. Melanoma tumor suppressor lncRNAs are highlighted in red.

**Figure 2 cells-11-00577-f002:**
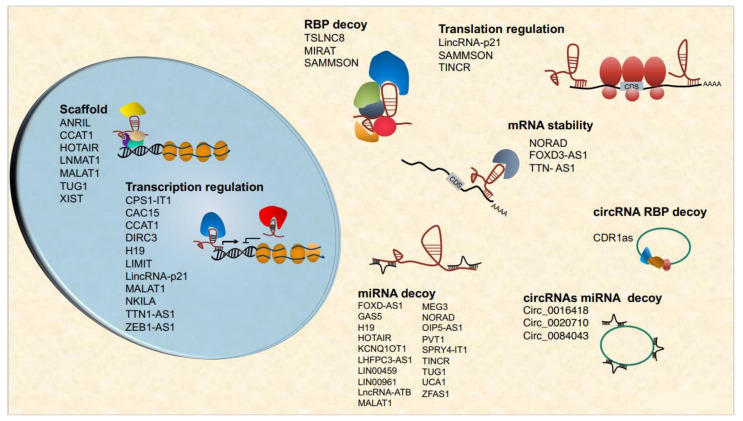
Major functional classes of nuclear and cytoplasmic lncRNAs involved in melanoma progression.

## Data Availability

Not applicable.

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
