# Peer review of "Regulation of LncRNAs in Melanoma and Their Functional Roles in the Metastatic Process"

_cells, 2022, doi:10.3390/cells11030577_

Round 1
Reviewer 1 Report
The manuscript “Regulation of lncRNAs in melanoma and their functional role in the metastatic process” by Melixetian et al reviews the role of long non-coding RNAs (lncRNAs) in melanoma.
After a general overview of both oncogenic and tumor suppressive lncRNAs, the authors then highlight several aspects of lncRNA function in more detail, such as their role in metastasis, immune evasion, drug resistance, as well as their potential utility as biomarkers and drug targets.
This is an overall well written manuscript that covers all important aspects of lncRNA regulation and function.
As a minor criticism, I would suggest to increase the font size of the lncRNA symbols in Fig. 1 by 1 or 2 points.
Author Response
We thank the Reviewer for the appreciation, we increased the font size of the lncRNA symbols in Fig. 1 by 2 points.
Reviewer 2 Report
Comments to authors:
Recent discoveries facilitated by next-generation sequencing technologies have revealed multiple roles of RNA beyond its conventional roles in protein synthesis. These non-coding RNAs play critical roles in almost all expects of gene expression regulation, modification of other RNAs, and formation of functional complexes with proteins and metabolites. Thus, many non-coding RNAs are implicated in many diseases, including various cancers. In this review, Melixetian et al. have presented a comprehensive discussion of different aspects of long-noncoding RNAs (lncRNAs) involved in melanoma and their potential implications as biomarkers and therapeutic targets.
Although some grammatical glitches need to be carefully proofread, the manuscript is well-written, comprehensive, and not difficult to follow. Overall, I think this review would be helpful for the ncRNA research community and beyond; therefore, the manuscript is worth publishing in a reputed journal like Cells. However, I have some comments that I expect authors would address before the manuscript could be accepted for a potential publication.
- Minor point, maybe “roles” rather than “role” sounds better in the title.
- Page 2, line 86, the “Nc-RNAs” should be written without the dash to make it consistent with other parts of the manuscript.
- Page 3, line 129, “lncRNA” should be “lncRNAs”.
- The authors presented a nice figure summarizing the lncRNAs involved in melanoma and their association with different biological processes. However, I think the manuscript would be more apparent and impressive if authors could also include a similar figure to illustrate the mechanisms by which these lncRNAs contribute to melanoma progression. For example, they may serve as protein sequesters, transporters or regulators, etc.
Author Response
We thank the reviewer for their comments and suggestions.
In the revised version of the manuscript we have:
According to the reviewer's suggestions, we have:
- corrected the title
- eliminated the dash on page 2, line 86
- corrected lncRNA with lncRNAs
- inserted a second figure to illustrate the mechanisms by which these lncRNAs contribute to melanoma progression, indicated as Figure 2 (please see the attachment).

Round 2
Reviewer 2 Report
Thanks to the authors for meticulously addressing my comments and concerns. Here are some minor points to revise in the final form of the manuscript before publication.
1) Page 3, last paragraph: first and second sentences start with "LncRNAs." It would be better to use a pronoun for the second sentence.
2) I did not find the Figure 2 citation in the manuscript text. Cite the figure in an appropriate place within the manuscript text.
Author Response
We thank the reviewer for the careful evaluation of the revised text. We have corrected it according to the suggestions. Revised version 2 is attached.
